# DoReMi: Optimizing Data Mixtures Speeds Up Language Model Pretraining

**Sang Michael Xie**[1,2], **Hieu Pham**[1], **Xuanyi Dong**[1], **Nan Du**[1], **Hanxiao Liu**[1], **Yifeng Lu**[1], **Percy Liang**[2], **Quoc V. Le**[1], **Tengyu Ma**[2], and **Adams Wei Yu**[1]

[1]Google DeepMind
[2]Stanford University

## Abstract

The mixture proportions of pretraining data domains (e.g., Wikipedia, books, web text) greatly affect language model (LM) performance. In this paper, we propose Domain Reweighting with Minimax Optimization (DoReMi), which first trains a small proxy model using group distributionally robust optimization (Group DRO) over domains to produce domain weights (mixture proportions) without knowledge of downstream tasks. We then resample a dataset with these domain weights and train a larger, full-sized model. In our experiments, we use DoReMi on a 280M-parameter proxy model to set the domain weights for training an 8B-parameter model (30x larger) more efficiently. On The Pile, DoReMi improves perplexity across *all* domains, even when it downweights a domain. DoReMi improves average few-shot downstream accuracy by 6.5% points over a baseline model trained using The Pile's default domain weights and reaches the baseline accuracy with 2.6x fewer training steps. On the GLaM dataset, DoReMi, which has no knowledge of downstream tasks, even matches the performance of using domain weights tuned on downstream tasks.

## 1 Introduction

Datasets for training language models (LMs) are typically sampled from a mixture of many domains [17; 13; 10; 9]. For example, The Pile [17], a large publicly available dataset, is composed of 24% web data, 9% Wikipedia, 4% GitHub, etc.[1] The composition of the pretraining data greatly affects the effectiveness of an LM [13; 20; 55]. However, it is unclear how much of each domain to include to produce a model that performs well for a wide variety of downstream tasks.

Existing works determine domain weights (the sampling probabilities for each domain) by using intuition or a set of downstream tasks. For example, The Pile uses heuristically-chosen domain weights, which could be suboptimal. On the other hand, existing LMs such as PaLM [10] and GLaM [13] tune the domain weights based on a set of downstream tasks, but requires training potentially thousands of LMs on different domain weights and risks overfitting to the particular set of downstream tasks.

Instead of optimizing domain weights based on a set of downstream tasks, our approach aims to find domain weights which lead to models that perform well on all domains by minimizing the worst-case *excess loss* over domains, following Oren et al. [35]; Mindermann et al. [30]. The excess loss is the loss gap between the model being evaluated and a pretrained reference model.

This motivates our algorithm, **Do**main **Re**weighting with **Mi**nimax Optimization (DoReMi), which leverages distributionally robust optimization (DRO) to tune the domain weights without knowledge of downstream tasks (Figure 1). First, DoReMi trains a small reference model (e.g., 280M parameters)

---

[1]The domain weights, which are based on token count in this paper, varies by tokenizer; see Appendix C.

37th Conference on Neural Information Processing Systems (NeurIPS 2023).

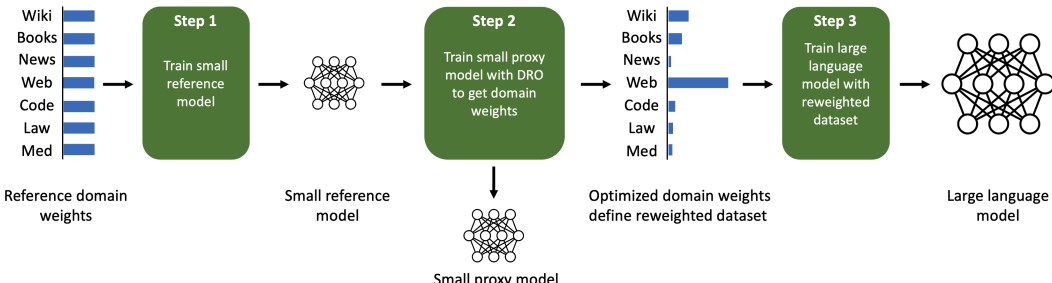

Figure 1: Given a dataset with a set of domains, Domain Reweighting with Minimax Optimization (DoReMi) optimizes the domain weights to improve language models trained on the dataset. First, DoReMi uses some initial reference domain weights to train a reference model (Step 1). The reference model is used to guide the training of a small proxy model using group distributionally robust optimization (Group DRO) over domains [35; 43; 34], which we adapt to output domain weights instead of a robust model (Step 2). We then use the tuned domain weights to train a large model (Step 3).

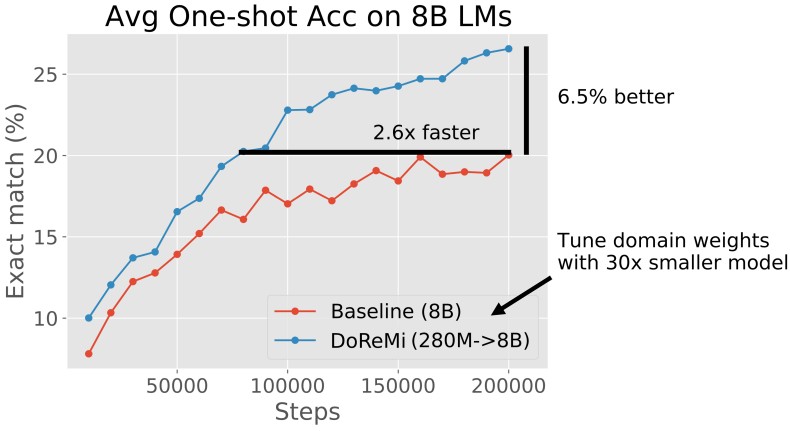

Figure 2: DoReMi optimizes domain weights with a small model (280M params) and uses these domain weights to train a much larger model (8B params, 30x larger). Here, optimizing the domain weights (training a small model twice) takes 8% of the compute of training the large model. DoReMi improves average one-shot downstream accuracy by 6.5% points and reaches the baseline accuracy 2.6x faster when pretraining on The Pile.

in a standard way. Second, DoReMi trains a small distributionally robust language model (DRO-LM) [35], which minimizes the worst-case excess loss (relative to the reference's model's loss) across all domains. Notably, *rather than using the robust LM*, we take the domain weights produced by DRO training. Finally, we train a large (8B) LM on a new dataset defined by these domain weights.

Our approach adapts the DRO-LM framework [35] to optimize domain weights instead of producing a robust model. To do this, DoReMi uses the online learning-based optimizer from Group DRO [43; 34], which dynamically updates domain weights according to the loss on each domain for rescaling the training objective, instead of sub-selecting examples from a minibatch as in Oren et al. [35]; Mindermann et al. [30]. Finally, DoReMi takes the averaged domain weights over DRO training steps.

In Section 3, we run DoReMi on 280M proxy and reference models to optimize domain weights on The Pile [17] and the GLaM dataset [13] (used in PaLM [10]). The DoReMi domain weights are used to train an 8B parameter LM (over 30x larger). On The Pile, DoReMi reduces perplexity on *all* domains over baseline domain weights, even when it downweights a domain. DoReMi improves average downstream accuracy over a baseline model trained on The Pile's default domain weights by 6.5% points on generative few-shot tasks and achieves the baseline downstream accuracy 2.6x faster (Figure 2). In Section 4, we find that DoReMi consistently improves LM training when varying the sizes of the proxy model and the main model trained with optimized domain weights. On the

GLaM dataset where domain weights tuned on downstream tasks are available, DoReMi even performs comparably to tuning domain weights on downstream task performance.[2]

## 2  Domain Reweighting with Minimax Optimization (DoReMi)

In this section we define DoReMi, an algorithm for using a small proxy model to optimize the domain weights of a language modeling dataset, which then improves the training of a large model.

**Setup.**  Suppose that we have $k$ domains (e.g., Wikipedia, GitHub), where for each domain $i$, we have a set of examples $D_i$. Domain weights $\alpha \in \Delta^k$ specify a probability distribution over the $k$ domains, and consequently a distribution over the training data: $P_\alpha = \sum_{i=1}^{k} \alpha_i \cdot \mathrm{unif}(D_i)$ where $\mathrm{unif}(D) = \frac{1}{|D|}\sum_{x \in D}\delta_x$ is the uniform distribution over the examples in $D$ and $\delta_x(x')$ is 1 if $x' = x$ and 0 otherwise.

**DoReMi.**  The inputs of DoReMi are the data $D_1, \ldots, D_k$, reference domain weights $\alpha_{\mathrm{ref}}$ (e.g., uniform or based on raw token count of each domain), and training hyperparameters for the large, full-size model (number of training steps $T$ and batch size $b$). DoReMi returns optimized domain weights $\bar{\alpha}$ and ultimately, a large model trained on $P_{\bar{\alpha}}$.

**Step 1: Obtain a small reference model.**  We first train a model $p_{\mathrm{ref}}$ on some reference domain weights $\alpha_{\mathrm{ref}}$ (e.g., based on raw token count as a default) for $T$ steps, batch size $b$. This model serves as the reference model for step 2 and captures a baseline level of difficulty of each example/domain. The reference model can be a relatively small model (280M parameters in our experiments).

**Step 2: Train proxy model with Group DRO to obtain domain weights.**  To obtain domain weights, we train a small *proxy model* $p_\theta$ in the distributionally robust language modeling (DRO-LM) [35] framework with the Group DRO optimizer [43], where $\theta$ are the weights of the proxy model. This framework trains a robust model by optimizing the worst-case loss over domains, which is equivalent to the following minimax objective:

$$\min_\theta \max_{\alpha \in \Delta^k} L(\theta, \alpha) := \sum_{i=1}^{k} \alpha_i \cdot \left[ \frac{1}{\sum_{x \in D_i}|x|} \sum_{x \in D_i} \ell_\theta(x) - \ell_{\mathrm{ref}}(x) \right] \qquad (1)$$

where the losses $\ell_\theta(x) = -\log p_\theta(x)$ and $\ell_{\mathrm{ref}}(x) = -\log p_{\mathrm{ref}}(x)$ are the negative log-likelihoods of the proxy and reference models respectively in this paper, and $|x|$ is the number of tokens in an example $x$. The objective aims to minimize the worst-case excess loss across domains because the inner maximization over $\alpha$ puts all the weight on the domain with the highest excess loss.

Intuitively, the excess loss $(\ell_\theta(x) - \ell_{\mathrm{ref}}(x))$ measures the headroom for the proxy model to improve, with respect to the reference model, on example $x$. Examples with higher excess loss are those where the reference model achieves low loss (such that the example is "learnable") but the proxy model still has high loss. Examples with low excess loss may be very high entropy (i.e. optimal loss is high, and thus the reference loss is high) or very low entropy (i.e., easy to learn, and thus the proxy loss is low). The Group DRO optimizer works by interleaving exponentiated gradient ascent updates on domain weights $\alpha_t$ with gradient updates on the proxy model weights $\theta_t$ over training steps $t$. The optimizer updates $\alpha_t$ to upweight domains with high excess loss, which scales up the proxy model's gradient update on examples from these domains. Following Nemirovski et al. [34], we return the average weights over the training trajectory $\bar{\alpha} = \frac{1}{T}\sum_{i=1}^{T}\alpha_t$ as the optimized domain weights to use in step 3.

**Step 3: Train large model with new domain weights.**  The tuned domain weights $\bar{\alpha}$ define a new training distribution $P_{\bar{\alpha}}$. We resample the data from this new distribution to train a main model (larger than the reference/proxy models), using a standard training procedure.

**Details for Step 2.**  Algorithm 1 provides the pseudocode for Step 2. The main structure of Algorithm 1 is a training loop which updates the proxy model over $T$ steps. At each step, we follow Sagawa et al. [43] and sample a minibatch with uniform domain weights (regardless of the reference domain

---

[2]A public re-implementation of DoReMi and optimized domain weights for The Pile can be found at https://github.com/sangmichaelxie/doremi.

---
**Algorithm 1** DoReMi domain reweighting (Step 2)
---
**Require:** Domain data $D_1,...,D_k$, number of training steps $T$, batch size $b$, step size $\eta$, smoothing parameter $c \in [0,1]$ (e.g., $c=$1e-3 in our implementation).
   Initialize proxy weights $\theta_0$
   Initialize domain weights $\alpha_0 = \frac{1}{k}\mathbf{1}$
   **for** $t$ from 1 to $T$ **do**
      Sample minibatch $B = \{x_1,...,x_j\}$ of size $b$ from $P_u$, where $u = \frac{1}{k}\mathbf{1}$
      Let $|x|$ be the token length of example $x$ ($|x| \leq L$)
      Compute per-domain excess losses for each domain $i \in \{1,2,...,k\}$ ($\ell_{\theta,j}(x)$ is $j$-th token-level loss):
$$\lambda_t[i] \leftarrow \frac{1}{\sum_{x \in B \cap D_i}|x|}\sum_{x \in B \cap D_i}\sum_{j=1}^{|x|}\max\{\ell_{\theta_{t-1},j}(x) - \ell_{\text{ref},j}(x),0\}$$
      Update domain weights (exp is entrywise): $\alpha_t' \leftarrow \alpha_{t-1}\exp(\eta\lambda_t)$
      Renormalize and smooth domain weights: $\alpha_t \leftarrow (1-c)\frac{\alpha_t'}{\sum_{i=1}^k \alpha_t'[i]} + cu$
      Update proxy model weights $\theta_t$ for the objective $L(\theta_{t-1},\alpha_t)$ (using Adam, Adafactor, etc.)
   **end for**
   **return** $\frac{1}{T}\sum_{t=1}^T \alpha_t$
---

weights $\alpha_{\text{ref}}$, which only affects the reference model). We then compute the per-domain excess losses, normalized by the total number of tokens in each domain, and use them to update the domain weights $\alpha_t$ at each step. We first compute the per-domain excess loss at a per-token level and then aggregate, where the token-level losses at index $j$ are $\ell_{\theta_{t-1},j}(x) = -\log p_{\theta_{t-1}}(x_j \,|\, x_1,...,x_{j-1})$ and $\ell_{\text{ref},j}(x) = -\log p_{\text{ref}}(x_j \,|\, x_1,...,x_{j-1})$. Since the Group DRO optimizer [43] requires a non-negative loss, we clip the per-token excess loss at 0. Finally, we update the proxy model for the objective $L(\theta_{t-1},\alpha_t)$ using a standard optimizer such as Adam [26] or Adafactor [46]. All experiments in this paper use Adafactor. We set the domain weight update step size to $\eta=1$ and the smoothing parameter to $c=$1e-3 in all our experiments and did not extensively tune these hyperparameters.

**Iterated DoReMi.** We extend DoReMi by running it for multiple rounds, setting the reference domain weights $\alpha_{\text{ref}}$ for the next round to be $\bar{\alpha}$ from the previous round. We call this *iterated DoReMi*. The entire iterated process still only uses small models for tuning domain weights. We stop iterating when the domain weights converge, which we define as when maximum change in any domain weight $\|\bar{\alpha} - \alpha_{\text{ref}}\|_\infty$ is less than 1e-3. Empirically, this takes only 3 rounds on the GLaM dataset (Section 3.2).

## 3 DoReMi Improves LM Training Efficiency and Performance

In this section, we use DoReMi domain weights optimized with a 280M-parameter proxy model to train a 8B-parameter main model (30x larger). We consider two datasets, The Pile [17] and the GLaM dataset [13]. On The Pile, DoReMi reduces perplexity significantly on every domain, improves average downstream accuracy on generative one-shot tasks by 6.5%, and achieves the baseline accuracy 2.6x faster. On the GLaM dataset where domain weights tuned on downstream datasets are available, DoReMi finds domain weights with comparable performance to downstream-tuned domain weights.

### 3.1 Experimental setup

**The Pile dataset.** The Pile [17] is a 800GB text dataset with 22 domains (Table 1). The default domain weights were determined heuristically. We use the default domain weights from The Pile dataset to train the baseline and as the reference domain weights $\alpha_{\text{ref}}$ in DoReMi (see Appendix C).

**GLaM dataset.** The GLaM dataset [13] (also used in training PaLM [10]) includes text from 8 domains (Table 2). For comparison, the GLaM domain weights (downstream-tuned) were tuned according to the downstream performance of models trained on each domain and the size of each domain [13]. We consider this an oracle comparison, since these domain weights are tuned on downstream tasks that are in our evaluation set. We use uniform domain weights both for training the baseline and the reference domain weights $\alpha_{\text{ref}}$ for DoReMi.

**Training setup.** We train Transformer [51] decoder-only LMs with the standard next-token language modeling loss. We conduct a controlled comparison by equalizing the amount of compute, measured by the number of tokens processed during training. For The Pile, we train each model for 200k steps; for the GLaM dataset, we train each model for 300k steps. All models use a batch size of 512 and maximum token length of 1024. The proxy and reference models have 280M parameters. All models are trained from scratch (other hyperparameters are in Appendix C).

**Evaluation.** We use held-out validation data to measure the perplexity on each domain. For downstream evaluation, we use the generative one-shot tasks from the GPT-3 paper [9]: TriviaQA [21], NaturalQuestions [27], WebQuestions [5], SQuADv2 [41], and LAMBADA [36]. We use the standard exact-match accuracy metric for the these datasets. The performance on these datasets (particularly TriviaQA) has been shown to correlate well with model scale even at the 100M–1B range [9].

**Compute used for optimizing domain weights.** We train two 280M models (the reference and proxy models) to optimize the domain weights. This is 8% of the FLOPs required to train the main 8B model. All FLOPs come from standard forward and backward passes.

**Notation for model sizes in DoReMi.** We denote the size of the reference/proxy models (which are always the same size in our experiments) and the size of the main model trained with DoReMi domain weights as "DoReMi (size of reference/proxy→size of main model)": for example, DoReMi (280M→8B). When we are discussing the optimized domain weights independently of the main model, we only include one number (e.g., DoReMi (280M)) which refers to the reference/proxy model size.

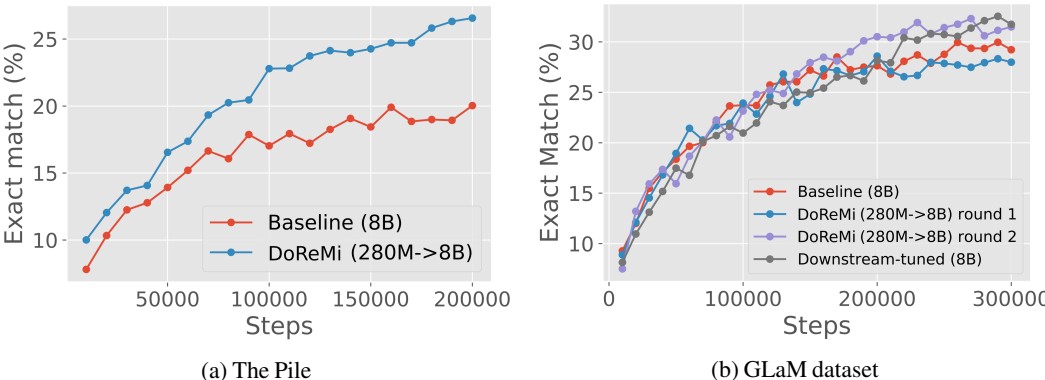

(a) The Pile        (b) GLaM dataset

Figure 3: Average one-shot downstream accuracy (exact match) on 5 tasks, with 8B parameter models trained on The Pile (left) and the GLaM dataset (right). On The Pile, DoReMi improves downstream accuracy by 6.5% points and achieves the baseline accuracy 2.6x faster (same plot as Figure 2). On the GLaM dataset, iterated DoReMi (round 2) attains comparable performance to oracle domain weights tuned with downstream tasks that are in our evaluation set.

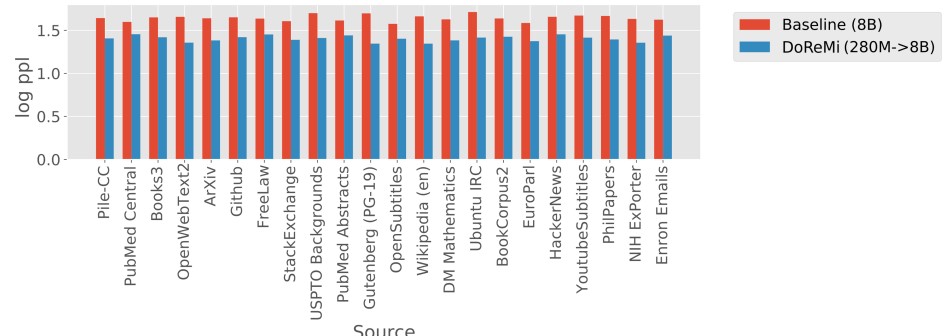

Figure 4: Per-domain log-perplexity of 8B models on The Pile. Despite downweighting some domains, DoReMi improves log-perplexity on all domains.

Table 1: Domain weights on The Pile. Baseline domain weights are computed from the default Pile dataset. DoReMi (280M) uses a 280M proxy model to optimize the domain weights.

| Domain | Baseline | DoReMi (280M) | Difference | Domain | Baseline | DoReMi (280M) | Difference |
|---|---|---|---|---|---|---|---|
| Pile-CC | 0.1121 | 0.6057 | +0.4936 | DM Mathematics | 0.0198 | 0.0018 | -0.0180 |
| YoutubeSubtitles | 0.0042 | 0.0502 | +0.0460 | Wikipedia (en) | 0.0919 | 0.0699 | -0.0220 |
| PhilPapers | 0.0027 | 0.0274 | +0.0247 | OpenWebText2 | 0.1247 | 0.1019 | -0.0228 |
| HackerNews | 0.0075 | 0.0134 | +0.0059 | Github | 0.0427 | 0.0179 | -0.0248 |
| Enron Emails | 0.0030 | 0.0070 | +0.0040 | FreeLaw | 0.0386 | 0.0043 | -0.0343 |
| EuroParl | 0.0043 | 0.0062 | +0.0019 | USPTO Backgrounds | 0.0420 | 0.0036 | -0.0384 |
| Ubuntu IRC | 0.0074 | 0.0093 | +0.0019 | Books3 | 0.0676 | 0.0224 | -0.0452 |
| BookCorpus2 | 0.0044 | 0.0061 | +0.0017 | PubMed Abstracts | 0.0845 | 0.0113 | -0.0732 |
| NIH ExPorter | 0.0052 | 0.0063 | +0.0011 | StackExchange | 0.0929 | 0.0153 | -0.0776 |
| OpenSubtitles | 0.0124 | 0.0047 | -0.0077 | ArXiv | 0.1052 | 0.0036 | -0.1016 |
| Gutenberg (PG-19) | 0.0199 | 0.0072 | -0.0127 | PubMed Central | 0.1071 | 0.0046 | -0.1025 |

Table 2: Domain weights in the GLaM dataset. Iterated DoReMi (280M) converges within 3 rounds, with a similar overall pattern to domain weights tuned on downstream tasks.

| | Round 1 | Round 2 | Round 3 | Downstream-tuned |
|---|---|---|---|---|
| Wikipedia | 0.09 | 0.05 | 0.05 | 0.06 |
| Filtered webpages | 0.44 | 0.51 | 0.51 | 0.42 |
| Conversations | 0.10 | 0.22 | 0.22 | 0.27 |
| Forums | 0.16 | 0.04 | 0.04 | 0.02 |
| Books | 0.11 | 0.17 | 0.17 | 0.20 |
| News | 0.10 | 0.02 | 0.02 | 0.02 |

## 3.2 DoReMi improves perplexity and downstream accuracy

We show that DoReMi significantly improves both the perplexity and downstream accuracy of 8B models trained on The Pile and the GLaM dataset over their respective baseline domain weights.

**Downstream accuracy improves on The Pile.** Figure 3 (left) shows the average downstream performance for baseline and DoReMi (280M→8B) models on The Pile. DoReMi improves the downstream accuracy by 6.5% points and achieves the baseline accuracy within 75k steps — 2.6x faster than the baseline (200k steps). Thus, DoReMi can dramatically speed up training and improve downstream performance.

**DoReMi can reduce perplexity across all domains without a tradeoff.** Figure 4 shows the per-domain log-perplexity of the 8B models on The Pile. DoReMi significantly reduces the perplexity over the baseline across *all* domains, despite allocating lower weight to some domains. How can this occur? One hypothesis is that the domains with the lowest and highest entropy can be downweighted without impacting the perplexity much. The lowest entropy domains statistically require few samples to learn. The highest entropy domains have token distributions that are close to common uniform priors — for example, models at random initialization tend to output a uniform next token distribution. Thus, we need less samples to fit these domains. Positive transfer from allocating more samples to medium entropy domains can then improve perplexity on all domains. In Appendix D, we provide a simple example where reweighting domains can improve perplexity on all domains and DoReMi finds such domain weights in simulations.

**Iterated DoReMi achieves performance of downstream-tuned weights on the GLaM dataset.** We employ iterated DoReMi on the GLaM dataset over 3 rounds. We find that the second and third round domain weights are almost identical (Table 2). Figure 3 (right) shows one-shot results for the first two rounds of iterated DoReMi. After the first round, the DoReMi main model has comparable downstream accuracy to the baseline (uniform domain weights). After the second round, the DoReMi main model achieves comparable downstream accuracy to oracle domain weights tuned on downstream tasks in our evaluation set. Overall, domain reweighting has a smaller effect on GLaM, possibly because there are only 8 domains compared to 22 in The Pile.

**Inspecting the DoReMi domain weights.** Tables 1 and 2 present the DoReMi domain weights for The Pile and the GLaM dataset. When running DoReMi on a 280M proxy model (DoReMi (280M)), most weight is put on the diverse Pile-CC web text domain. Note that Wikipedia is downweighted in comparison to the baseline, but DoReMi still improves the downstream accuracy on tasks derived from Wikipedia (e.g., TriviaQA, Appendix Table 5). Domain weights for a 1B proxy model (Appendix 8)

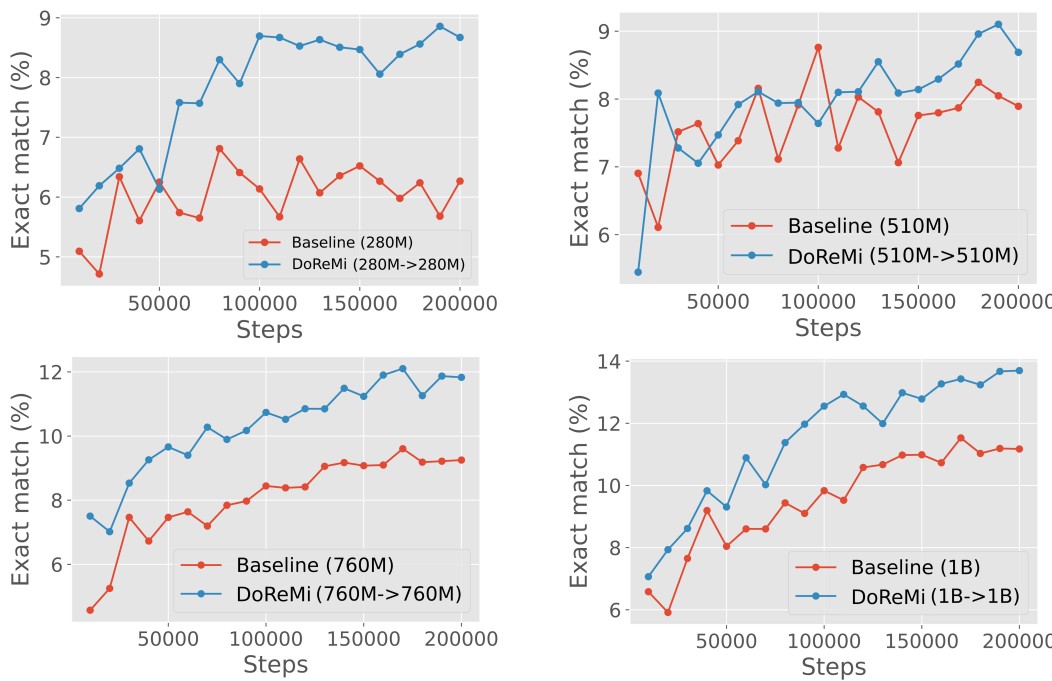

Figure 5: Average one-shot downstream accuracy across 4 model scales (280M, 510M, 760M, 1B) where the reference/proxy models for DoReMi are the same size as the main model trained with DoReMi domain weights. DoReMi consistently improves downstream accuracy across scales, with a similar 3% accuracy gap at 200k steps at most scales (except for 510M). DoReMi achieves the baseline accuracy 4x faster on average across scales.

shows a different trend, where OpenWebText is the mostly upweighted instead of Pile-CC. This suggests that there may be multiple possible local minima in the domain weight space. On the GLaM dataset, the DoReMi weights have the same general pattern as the downstream-tuned domain weights. DoReMi is able to recover a similar set of domain weights by starting from uniform initial reference domain weights, without any use of downstream data.

## 4 Ablations and Analysis Across Scales

Previously in Section 3, we showed that DoReMi finds domain weights using 280M models that can improve training of 8B models. In this section, we conduct an analysis of DoReMi where we vary the scale of the proxy model in relation to the main model and ablate the components of the excess loss objective.

**DoReMi improves LMs consistently across scales.** We consider using proxy and main models of the same size to analyze DoReMi's behavior in a simple setting, without the need for the domain weights to generalize across scales. Note that this is just for scientific purposes since this does not save compute in practice. In particular, we run DoReMi (X→X) where X is 280M, 510M, 760M, or 1B on The Pile. Figure 5 shows that DoReMi consistently improves downstream accuracy over the baseline by 2% and achieves the baseline accuracy 4x faster on average across scales, and this improvement does not shrink with larger model size. DoReMi improves the worst-case perplexity on all scales and improves 18 of 22 individual domain perplexities on average across scales (Appendix Table 6). These experiments give a rough picture of how much is lost when using a smaller proxy model; our DoReMi (280M→8B) model achieves the baseline accuracy 2.6x faster, while matching the proxy and main model sizes results in a 4x average speedup.

**Proxy model underperforms main model, especially at larger sizes.** Recall that DoReMi uses Group DRO to train a proxy model, which reweights the objective with the domain weights. In contrast, the main model is trained by resampling on the domain weights from DoReMi. When the proxy model and the main model are the same size, which one is the better model? Table 3b shows that the proxy

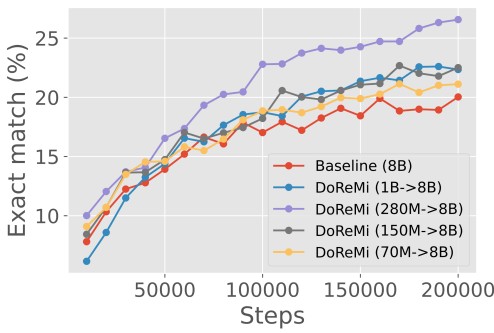 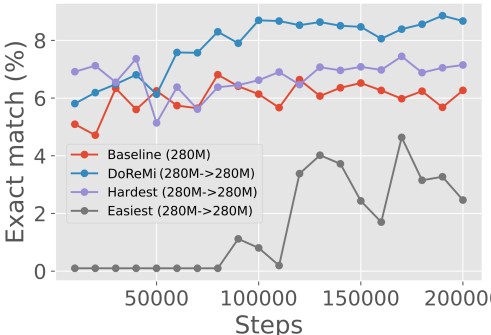

Figure 6: Average downstream accuracy for models trained on The Pile. **(Left)** Increasing the size of the reference/proxy models from 70M to 280M in DoReMi improves downstream accuracy for a 8B main model, but the trend does not continue for the 1B proxy model. We hypothesize that the Group DRO optimizer is worse for larger proxy models. **Right)** Optimizing for the hardest or easiest domains rather than excess loss (which combines both) do not achieve the same average downstream accuracy as DoReMi (280M models).

Table 3: Summary of per-domain log-perplexities on The Pile (22 total domains). Average log-perplexity is an unweighted average of the per-domain log-perplexities.

(a) Varying the size of the proxy/reference model and training at 8B.

| | Worst-case log-ppl | Avg log-ppl | # domains beating baseline |
|---|---|---|---|
| Baseline (8B) | 1.71 | 1.64 | 0/22 |
| DoReMi (70M->8B) | 1.63 | 1.53 | 22/22 |
| DoReMi (150M->8B) | 1.56 | 1.52 | 22/22 |
| DoReMi (280M->8B) | 1.46 | 1.40 | 22/22 |
| DoReMi (1B->8B) | 1.58 | 1.54 | 22/22 |

(b) Perplexity of the DoReMi main model and proxy model of the same size. Although the 1B proxy model is relatively poor quality, the resulting domain weights still improve the main model.

| | Worst-case log-ppl | Avg log-ppl | # domains beating baseline |
|---|---|---|---|
| Baseline (280M) | 2.39 | 2.32 | 0/22 |
| DoReMi (280M->280M) | 2.19 | 2.13 | 22/22 |
| Proxy (280M) | 2.33 | 2.27 | 19/22 |
| Baseline (1B) | 1.94 | 1.87 | 0/22 |
| DoReMi (1B->1B) | 1.92 | 1.83 | 19/22 |
| Proxy (1B) | 2.11 | 2.02 | 0/22 |

model typically underperforms the main model in this case. The gap between the proxy and main model increases with scale, as the 1B proxy model not only underperforms the 1B main model but also the 1B baseline model, while the 280M proxy model achieves better perplexity than the 280M baseline model on 19/22 domains. Despite the relatively poor quality of the 1B proxy model, the domain weights still allow the 1B main model to achieve the baseline performance over 2x faster. This suggests that DoReMi can succeed even if the proxy model is not trained well. However, we hypothesize that the mismatch between the proxy and main model training (loss reweighting vs. resampling) explains their performance difference and therefore a resampling-based Group DRO optimizer may improve DoReMi for larger proxy models.

**Effect of proxy model scale on larger main model's performance.** We consider 70M, 150M, 280M, and 1B scales for the DoReMi proxy model while fixing the main model size at 8B (DoReMi (X→8B)). From 70M to 280M, increasing the proxy model size improves downstream accuracy at 8B (Figure 6 left). We hypothesize that this trend does not continue for the 1B proxy model because the Group DRO optimizer is worse at larger scales (Table 3b). While DoReMi (280M→8B) results in the most improvement at 8B, DoReMi (150M→8B) and DoReMi (1B→8B) still achieve the baseline accuracy almost 2x faster. This suggests that DoReMi is robust to the proxy model scale. In practice, we suggest choosing a relatively small proxy model size (280M) to save compute.

**Choosing the easiest or hardest domains do not suffice.** We ablate the components of the excess loss metric $\ell_\theta(x) - \ell_{\text{ref}}(x)$ by running DoReMi using only the loss of the proxy model $p_\theta$ on example $x$, i.e. $\ell_\theta(x)$ (prefer hardest domains for the proxy model) or only the negative loss of the reference $-\ell_{\text{ref}}(x)$ (prefer easiest domains for the reference model). Figure 6 (right) shows that neither of the components of the excess loss alone are sufficient to achieve the gains of DoReMi.

# 5   Related Work

**Curating pretraining data for LMs.**   Most closely related is the GLaM dataset [13] (also used for training PaLM [10]), which has domain weights that are tuned using downstream data. Optimizing domain weights for downstream tasks can be expensive and could require search/zero-order optimization [48], RL [56], or heuristic assumptions on how positive/negative transfer between domains work. Example-level filtering also brings benefits for LM training. The C4 dataset [39] shows gains over CommonCrawl via heuristic data cleaning methods. Du et al. [13]; Xie et al. [55] show that filtering the data at an example level for high-quality text that look like Wikipedia and books can significantly improve downstream performance for LMs. In contrast to these works, DoReMi sets domain weights automatically with only two small LM training runs and does not make assumptions about the type of data to prefer (Wikipedia-like, etc.).

**General data selection methods.**   Moore-Lewis selection [32; 3; 15] selects examples with high cross-entropy difference (similar to excess log-perplexity) between language models trained on target and raw data. In contrast, DoReMi reweights the data without a target distribution. Coleman et al. [11] select examples based on the uncertainty of a small proxy model for active learning, while DoReMi uses DRO on the excess loss with respect to a reference model, and focuses on data mixture reweighting. Mindermann et al. [30] select examples in an online fashion by taking the top $k$ examples in a minibatch according to excess loss. DoReMi optimizes the data mixture before training, allowing the larger main model to train in a standard way. Many other works on data selection are in vision [49; 22; 24; 23; 25; 53; 54; 38; 31; 45] and mainly focus on example-level subset selection with metrics such as gradient matching. Overall, these methods do not address data selection for pretraining, where the downstream data distribution may be very different from the pretraining distribution. DoReMi aims to address the pretraining/downstream distribution shift with a robust optimization approach. To the best of our knowledge, we are the first to show that reweighting the data according to losses of a small proxy LM can improve the training efficiency of much larger LM.

**Distributionally robust optimization.**   Within DRO methods for deep learning [4; 47; 35; 43], we target a restricted form of shift called group shifts [14; 35; 43], where the test distribution can be an unknown mixture of groups (domains). We follow DRO-LM [35], which employs DRO for LMs in the group shift setting. DRO-LM also uses a baselined loss, but with a simple bigram reference model. DoReMi uses a reference model of the same size and architecture as the proxy model to ensure that the losses are on a similar scale. During optimization, DRO-LM takes a worst-case subset of each minibatch to update the model on, while we use the Group DRO optimizer [43] which doesn't require online subselection. If we equalize the number of examples in each minibatch used for gradient updates, online subselelction is more expensive than Group DRO since it requires running forward passes on a larger minibatch (e.g., double the minibatch size) before selecting a subset to update the model with. In comparison, the Group DRO optimizer updates the model on all examples in a weighted fashion. Overall, in contrast to these DRO methods which aim to produce robust **models**, we use DRO to optimize the **data** for training larger models more efficiently.

**Data-centric AI.**   Large-scale datasets and benchmarks have driven much of the recent progress in AI, including vision, NLP, and multimodal models [12; 42; 52; 40; 39; 17; 44; 16]. However, most datasets are still painstakingly created with human-generated data, manual work, and heuristics [12; 39; 17; 44; 16]. DoReMi is a principled data-centric method that aims to improve language model training efficiency. We hope that DoReMi can provide a starting point for a general data-centric framework for language modeling via robust optimization.

# 6   Discussion and Limitations

**Saving compute in DoReMi with extrapolation.**   In Section 2, we run DoReMi for the number of training steps that will be used to train the final model, which could be unnecessarily expensive. A future direction for saving compute would be to stop running DoReMi at an early step and extrapolate the domain weights for the desired number of steps, since we found that most of the variation in the domain weights during a DoReMi run seems to occur in the beginning of training (Appendix Figure 8).

**Choice of reference model.** The choice of reference model can affect the domain weights found by DoReMi. For example, iterated DoReMi (Section 3) improves performance by using a reference model trained on the tuned domain weights from a previous round of DoReMi. Further directions include varying the reference model size and using specialized reference models to optimize domain weights for a specific application area.

**What is a domain?** We define a domain by data provenance in our experiments, but this only enables coarse-grained control. Using fine-grained domains could improve the gains from DoReMi. For example, DoReMi is more effective on The Pile (22 domains) than the GLaM dataset (8 domains). Open directions include automatically finding fine-grained domains (e.g., via clustering as in DRO-LM [35]) and reweighting the data at an example level. When domains are very fine-grained, it will be important to control the pessimism of DRO (e.g., DRO can put all the weight on a small set of worst-case examples).

**Transferability of domain weights across scales.** We optimized the domain weights with a small proxy model (280M) and directly used these domain weights to improve training at a larger scale (8B). Understanding why the domain weights can be transferred across scales and the limits of how far these domain weights transfer are important questions to answer in future work.

**Broader impacts.** Large language models are We hope to improve training efficiency and reduce the environmental impact of training large LMs [50; 28; 37; 29]. In particular, by reducing the training time by 2x, we can halve the cost and energy consumption of training large language models. Since such efficiency improvements may be used to develop even larger models, there may be no absolute improvement in energy consumption. Ultimately, we hope to improve the training efficiency and cost of developing future language models relative to existing methods.

Large LMs have also been well-documented to have risks and biases [1; 33; 7; 6; 18]. For example, GPT-3 tends to have an anti-Muslim bias, where Muslims are frequently related to violence or terrorism in analogy and completion tasks [1]. As large language models are increasingly relied upon in applications, the magnitude of the risks increases [8]. Distributionally robust optimization (DRO), which is used in DoReMi to optimize the data mixture, can have a favorable impact on fairness [19]. While the standard approach of minimizing the average loss can lead to disparate performance on minority subgroups that do not contribute heavily to the loss [2], DRO promotes good performance on all groups via a worst-case loss. In this way, DRO-style data-centric methods such as DoReMi can improve the representation disparity between majority and minority subgroups in a dataset.

## 7 Conclusion

We introduced DoReMi, an algorithm reweighting data domains for training language models. DoReMi is able to run on small models and transfer the benefits to 30x larger models, resulting in a 2.6x speedup in training on the Pile just by changing the sampling probabilities on domains. We hope to instigate more research on data-centric approaches for improving language model training efficiency.

## Acknowledgments

We thank Xiangning Chen, Andrew Dai, Zoubin Ghahramani, Balaji Lakshminarayanan, Paul Michel, Yonghui Wu, Steven Zheng, Chen Zhu, anonymous reviewers, and the broader Google Bard team members for insightful discussions and pointers.

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

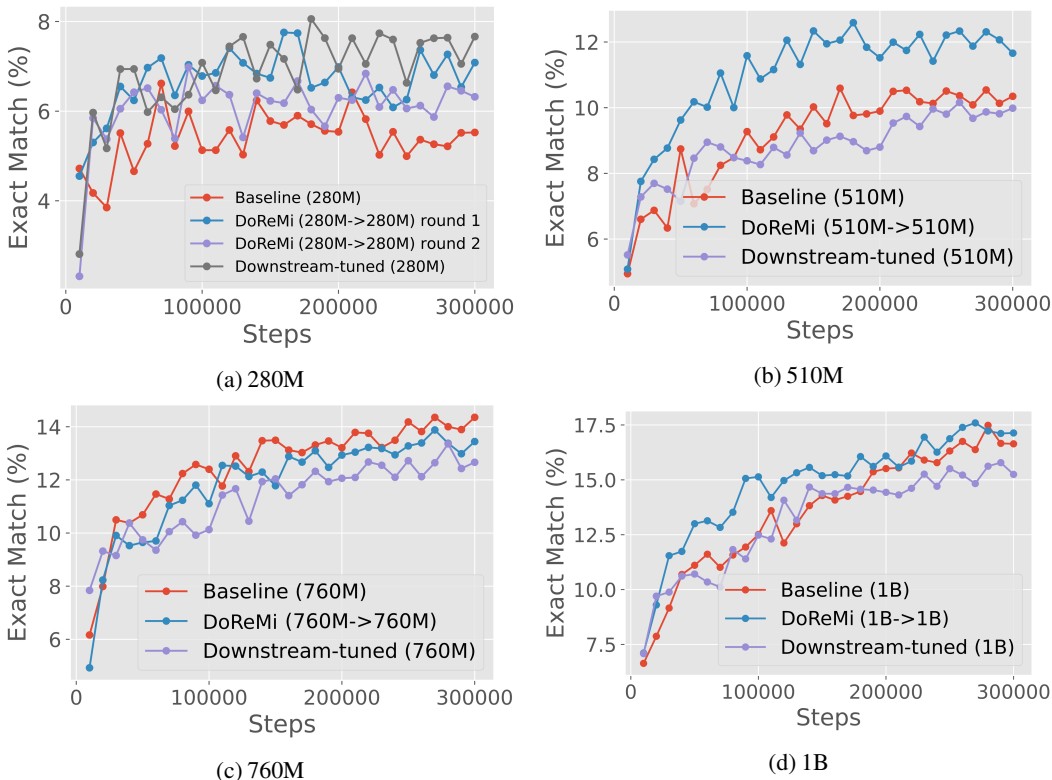

Figure 7: Average one-shot downstream accuracy across 4 model scales, where the reference/proxy models for DoReMi are the same size as the final model trained with DoReMi domain weights. All models in this figure are trained on the GLaM dataset. DoReMi consistently improves downstream accuracy across scales.

## A    Results Across Scales on the GLaM dataset

Figure 7 presents results across different scales (280M, 510M, 760M, 1B) on the GLaM dataset, where the proxy/reference models are the same size as the main model trained with DoReMi domain weights. Across all scales, DoReMi is comparable or better than both the baseline (uniform) domain weights and downstream-tuned domain weights. Interestingly, for iterated DoReMi at the 280M scale, the second round weights achieve slightly worse downstream accuracy than the round 1 weights when used to train 280M models, but transfer better to training 8B models.

## B    Detailed Results for The Pile

**Per-domain perplexities for 8B models.**   Table 4 shows per-domain perplexities for 8B models trained on the Pile. The reference/proxy models in this case are 70M, 150M, 280M, and 1B. DoReMi improves the perplexity on each domain compared to the baseline domain weights.

**Per-task accuracies for 8B models.**   Table 5 shows the accuracies on one-shot generative tasks for various reference/proxy model sizes from 70M to 1B. All DoReMi models improve downstream performance significantly over the baseline.

**Summary of perplexity results across scales.**   Table 6 shows a summary of per-domain perplexities for DoReMi across 4 scales (280M, 510M, 760M, 1B). Here, the reference/proxy models are the same size as the main model trained with DoReMi domain weights. On average, DoReMi improves perplexity on 18.25 out of 22 domains from The Pile. The worst-case perplexity is always reduced (or comparable in the 510M case) with respect to the baseline domain weights.

Table 4: Per-domain log-perplexities for 8B models trained on The Pile where the reference/proxy models are or smaller sizes (70M, 150M, 280M, 1B). Models trained with DoReMi domain weights have lower perplexity on all domains than the baseline weights.

| | Baseline (8B) | DoReMi (70M->8B) | DoReMi (150M->8B) | DoReMi (280M->8B) | DoReMi (1B->8B) |
|---|---|---|---|---|---|
| Pile-CC | 1.64 | 1.51 | 1.48 | 1.41 | 1.55 |
| PubMed Central | 1.60 | 1.58 | 1.54 | 1.46 | 1.56 |
| Books3 | 1.65 | 1.52 | 1.50 | 1.42 | 1.57 |
| OpenWebText2 | 1.66 | 1.48 | 1.54 | 1.36 | 1.58 |
| ArXiv | 1.64 | 1.56 | 1.53 | 1.38 | 1.51 |
| Github | 1.65 | 1.55 | 1.54 | 1.42 | 1.53 |
| FreeLaw | 1.64 | 1.55 | 1.54 | 1.45 | 1.55 |
| StackExchange | 1.61 | 1.52 | 1.54 | 1.39 | 1.55 |
| USPTO Backgrounds | 1.70 | 1.53 | 1.50 | 1.41 | 1.56 |
| PubMed Abstracts | 1.61 | 1.56 | 1.51 | 1.44 | 1.55 |
| Gutenberg (PG-19) | 1.70 | 1.56 | 1.54 | 1.35 | 1.52 |
| OpenSubtitles | 1.58 | 1.56 | 1.52 | 1.40 | 1.55 |
| Wikipedia (en) | 1.66 | 1.49 | 1.53 | 1.35 | 1.56 |
| DM Mathematics | 1.63 | 1.50 | 1.56 | 1.38 | 1.48 |
| Ubuntu IRC | 1.71 | 1.53 | 1.49 | 1.42 | 1.48 |
| BookCorpus2 | 1.64 | 1.57 | 1.54 | 1.43 | 1.57 |
| EuroParl | 1.59 | 1.52 | 1.51 | 1.37 | 1.53 |
| HackerNews | 1.66 | 1.50 | 1.55 | 1.45 | 1.55 |
| YoutubeSubtitles | 1.67 | 1.63 | 1.55 | 1.42 | 1.53 |
| PhilPapers | 1.67 | 1.55 | 1.49 | 1.39 | 1.53 |
| NIH ExPorter | 1.63 | 1.51 | 1.48 | 1.36 | 1.52 |
| Enron Emails | 1.62 | 1.48 | 1.52 | 1.44 | 1.56 |

Table 5: Per-task exact-match accuracies for generative one-shot tasks. All DoReMi models improve downstream performance significantly over the baseline domain weights.

| | Baseline | DoReMi (1B->8B) | DoReMi (280M->8B) | DoReMi (150M->8B) | DoReMi (70M->8B) |
|---|---|---|---|---|---|
| LAMBADA | 20.10 | 22.55 | 29.19 | 20.59 | 26.20 |
| NaturalQuestions | 4.35 | 6.01 | 7.73 | 6.26 | 5.10 |
| SQuADv2 | 44.43 | 42.22 | 51.89 | 46.53 | 40.99 |
| TriviaQA | 24.55 | 32.25 | 34.86 | 30.01 | 26.30 |
| WebQuestions | 6.74 | 8.71 | 9.15 | 9.15 | 6.99 |
| Average | 20.03 | 22.35 | 26.56 | 22.51 | 21.11 |

Table 6: Summary of per-domain log-perplexities for 280M, 510M, 760M, and 1B models trained on The Pile, where the reference/proxy models are the same size. DoReMi improves the worst-case and average perplexity of the baseline domain weights in all cases. On average, DoReMi improves perplexity on 18 out of 22 domains.

| | Worst-case log-ppl | Avg log-ppl | # domains beating baseline |
|---|---|---|---|
| Baseline (280M) | 2.39 | 2.32 | 0/22 |
| DoReMi (280M->280M) | 2.19 | 2.13 | 22/22 |
| Proxy (280M) | 2.33 | 2.27 | 19/22 |
| Baseline (510M) | 2.14 | 2.08 | 0/22 |
| DoReMi (510M->510M) | 2.14 | 2.06 | 15/22 |
| Proxy (510M) | 2.23 | 2.18 | 0/22 |
| Baseline (760M) | 2.05 | 1.97 | 0/22 |
| DoReMi (760M->760M) | 2.00 | 1.94 | 17/22 |
| Proxy (760M) | 2.15 | 2.10 | 0/22 |
| Baseline (1B) | 1.94 | 1.87 | 0/22 |
| DoReMi (1B->1B) | 1.92 | 1.83 | 19/22 |
| Proxy (1B) | 2.11 | 2.02 | 0/22 |

Table 7: Summary of perplexity results for ablations on the DRO objective (excess loss). The individual components (which prefer hardest and easiest domains respectively) do not reduce perplexity over the baseline.

| | Worst-case log-ppl | Avg log-ppl | # domains beating baseline |
|---|---|---|---|
| Baseline (280M) | 2.39 | 2.32 | 0 |
| DoReMi (280M->280M) | 2.19 | 2.13 | 22/22 |
| Hardest (280M->280M) | 2.66 | 2.62 | 0/22 |
| Easiest (280M->280M) | 4.27 | 4.18 | 0/22 |

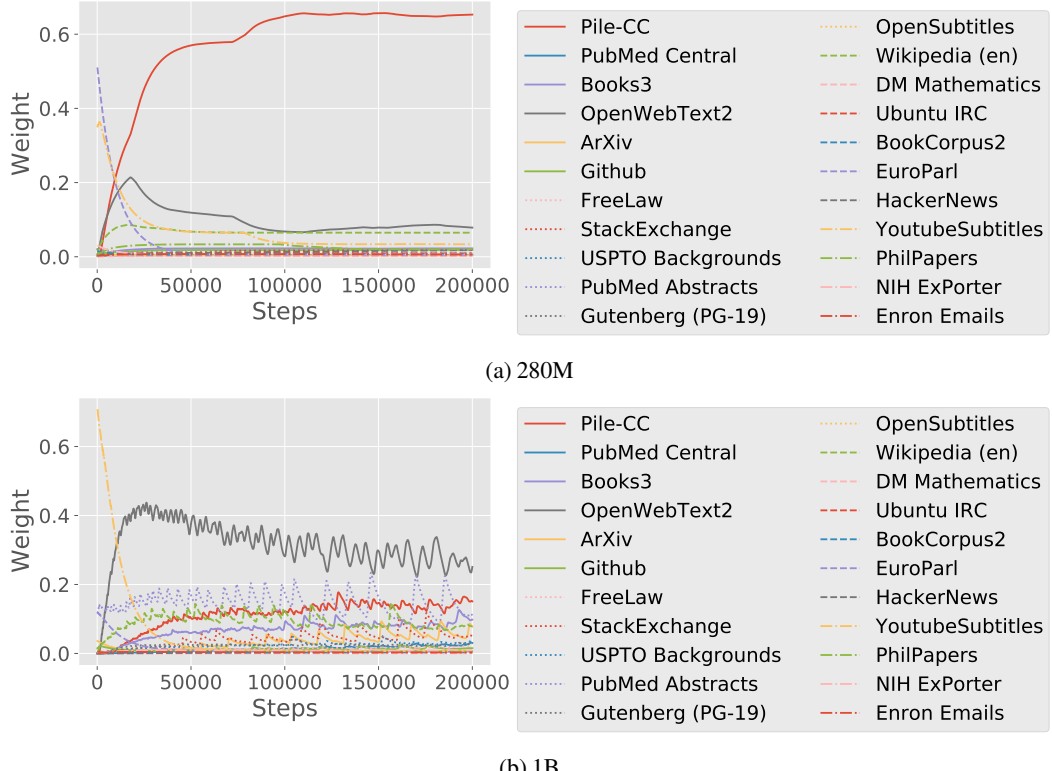

(a) 280M

(b) 1B

Figure 8: Exponential moving average of domain weights throughout a DoReMi run for 280M and 1B reference/proxy models. In the beginning of the run, the domain weights change quickly and then become more stable after 50k steps. This suggests that 1) smaller compute budgets may require drastically different domain weights, and 2) we may be able to save compute by extrapolating the domain weights after 50k steps.

**Perplexity results for ablations.** Table 7 shows the perplexities for ablations on the DRO objective. We change the DRO objective and use these to tune domain weights on 280M reference/proxy models. These tuned domain weights are then used to train a main 280M model. Hardest refers to optimizing the domain-level log-perplexity without baselining with a reference model. Easiest refers to optimizing for the domains with lowest log-perplexity under the reference model. Both ablations do not improve perplexity on any domain over the baseline. Optimizing for the "hardest" domain does not actually result in improving worst-case perplexity, supporting the results of Oren et al. [35], which also employs DRO for language modeling with a baselined loss.

**Trajectory of domain weights.** Figure 8 shows the exponential moving average (smoothing parameter 0.99) of domain weights during a run of DoReMi. In both cases, there are domains with very high weight initially and decrease in weight very quickly (within 50k steps). Since we compute the final domain weights by integrating these curves over steps and normalizing, this suggests that if we have a smaller compute budget, these domains could become more important — this highlights the dependence of the mixture weights on the compute budget. At the same time, the domain weights tend to quickly stabilize after 50k steps, suggesting that the optimal domain weights should be similar for larger compute budgets. We may also be able to take advantage of this stability after 50k steps to run DoReMi for a smaller number of steps and extrapolate the domain weights to save compute.

**Comparison of domain weights for 280M and 1B.** Table 8 presents the DoReMi domain weights for The Pile at 280M and 1B proxy models. Different proxy model sizes can result in different domain weights, which suggests that there may be multiple local minima in domain weight space. With a 280M proxy model, most of the weight is put on the Pile-CC web text domain, while DoReMi with a

Table 8: Domain weights on The Pile. Baseline domain weights are computed from the default Pile dataset. With different proxy model sizes, DoReMi (280M) and DoReMi (1B) result in different domain weights. Despite the differences, the qualitative patterns are similar other than the which web domain has the most weight.

| | Baseline | DoReMi (280M) | DoReMi (1B) |
|---|---|---|---|
| Pile-CC | 0.1121 | 0.6057 | 0.1199 |
| PubMed Central | 0.1071 | 0.0046 | 0.0149 |
| Books3 | 0.0676 | 0.0224 | 0.0739 |
| OpenWebText2 | 0.1247 | 0.1019 | 0.3289 |
| ArXiv | 0.1052 | 0.0036 | 0.0384 |
| Github | 0.0427 | 0.0179 | 0.0129 |
| FreeLaw | 0.0386 | 0.0043 | 0.0148 |
| StackExchange | 0.0929 | 0.0153 | 0.0452 |
| USPTO Backgrounds | 0.0420 | 0.0036 | 0.0260 |
| PubMed Abstracts | 0.0845 | 0.0113 | 0.1461 |
| Gutenberg (PG-19) | 0.0199 | 0.0072 | 0.0250 |
| OpenSubtitles | 0.0124 | 0.0047 | 0.0017 |
| Wikipedia (en) | 0.0919 | 0.0699 | 0.0962 |
| DM Mathematics | 0.0198 | 0.0018 | 0.0004 |
| Ubuntu IRC | 0.0074 | 0.0093 | 0.0044 |
| BookCorpus2 | 0.0044 | 0.0061 | 0.0029 |
| EuroParl | 0.0043 | 0.0062 | 0.0078 |
| HackerNews | 0.0075 | 0.0134 | 0.0058 |
| YoutubeSubtitles | 0.0042 | 0.0502 | 0.0159 |
| PhilPapers | 0.0027 | 0.0274 | 0.0063 |
| NIH ExPorter | 0.0052 | 0.0063 | 0.0094 |
| Enron Emails | 0.0030 | 0.0070 | 0.0033 |

1B proxy model puts most of the weight on OpenWebText2. The overall pattern of the domain weights for the rest of the domains are similar.

## C  Training Details

**Data preprocessing.**  For all datasets, we preprocessed the data by chunking into length 1024 examples with respect to a SentencePiece tokenizer with 256k vocabulary size. The examples are separated by domain to facilitate hierarchical sampling (first sample a domain according to some domain weights, then sample an example from that domain at random). To reduce the amount of padding tokens, we made an effort to pack examples (possibly from different domains) together into the same sequence. When doing such a packing, we compute the domain perplexities on a per-token level in DoReMi.

**Baseline domain weights for The Pile.**  The baseline domain weights for The Pile were computed from The Pile dataset and the number of epochs for each domain given in Gao et al. [17]. After chunking into length 1024 examples, we counted the number of examples in each domain and multiplied by the number of epochs that domain specified in Gao et al. [17]. We then normalized these counts to obtain the baseline domain weights.

**Training setup.**  For all training runs (including DRO runs), we train with a batch size of 512, initial learning rate of 1e-3, weight decay of 1e-2, and gradient clipping to norm 1. We decay the learning rate exponentially until it reaches a minimum of 1e-4 at the end of training, with a linear warmup of 6% of the total training steps. We train for 200k steps on The Pile and 300k steps on the GLaM dataset. Models under 1B parameters were trained with TPUv3 accelerators, while 1B and 8B models were trained with TPUv4.

**Model architectures.**  Table 9 shows the architecture hyperparameters for the model sizes used in the paper. All the models we use are vanilla Transformer decoder-only models with a 256k vocab size.

## D  Simple Example Where Data Reweighting Has No Tradeoff

Motivated by the findings in Section 3.2, we present a simple language modeling example where reweighting the training data from different domains improves perplexity on all domains. The example shows that DoReMi downweights domains that are extremely high or low entropy.

**Setup.**  Suppose the ground-truth distribution of text $p^*$ is a mixture over $k$ domains, where each domain $z \in \{1,...,k\}$ is defined by a different unigram distribution $p^*(x \mid z)$ over $m$ tokens. Given a

Table 9: Architecture hyperparameters for various model scales used in the paper. All models are vanilla Transformer decoder-only models and use vocabulary size 256k.

|  | Layers | Attention heads | Attention head dim | Model dim | Hidden dim |
|---|---|---|---|---|---|
| 70M | 3 | 4 | 64 | 256 | 1024 |
| 150M | 6 | 8 | 64 | 512 | 2048 |
| 280M | 12 | 12 | 64 | 768 | 3072 |
| 510M | 12 | 16 | 64 | 1024 | 8192 |
| 760M | 12 | 20 | 64 | 1280 | 8192 |
| 1B | 16 | 32 | 64 | 2048 | 8192 |
| 8B | 32 | 32 | 128 | 4096 | 24576 |

budget of $n$ training samples, the goal is choose domain weights $p(z)$ ($k$ scalars that add to 1) to sample training data with such that we learn the parameters of the unigram distributions $p^*(\cdot \,|\, z)$ well for all $z$ from 1 to $k$. Notably, we do not aim to estimate the ground truth mixture proportions across domains.

**Data.** Given some domain weights $p(z)$, we sample training data hierarchically: first we determine the number of samples $n_z$ per domain $z$ by drawing from a multinomial distribution over $k$ possibilities with probabilities defined by $p(z)$ and $n$ total trials. Then, for each domain $z$, we sample $n_z$ tokens from $p^*(\cdot \,|\, z)$, forming a vector of tokens $X_z$ with length $n_z$.

**Model.** For each domain $z$, we consider a Bayesian model of the unigram distribution $p(x \,|\, z;\theta)$ with a Dirichlet prior $p(\theta \,|\, z;\beta)$ over the unigram distribution parameters $\theta \in \Delta^m$. The Dirichlet prior has hyperparameters $\beta \in \mathbb{R}^m$, which can be viewed as a "pseudo-count" for each token. For each domain $z$, we estimate the parameters $\hat{\theta}_z$ by computing the mean of the posterior distribution conditioned on the data:

$$\hat{\theta}_z(x) = \frac{1}{n_z + s_z}\left[\lambda_z(x) + \sum_{i=1}^{n_z}\mathbf{1}[X_z[i] = x]\right] \quad \text{for all } x \in \{1,...,m\} \tag{2}$$

where $s_z = \sum_x \lambda_z(x)$ is the sum of pseudocounts.

For a domain $z$, we can write the parameter error of this estimator as a function of the "difficulty" $H_z$ of predicting the next token and the "quality" of the prior $\Delta_z$, defined below.

**Lemma D.1.** *For domain index $z$ with $n_z$ samples, the parameter error is*

$$\sum_x \mathbb{E}[(\hat{\theta}_z(x) - p^*(x \,|\, z))^2] = \frac{n_z H_z + s_z^2 \Delta_z}{(n_z + s_z)^2} \tag{3}$$

*where*

$$H_z = \sum_x p^*(x \,|\, z)(1 - p^*(x \,|\, z)) \tag{4}$$

$$\Delta_z = \sum_x \left(p^*(x \,|\, z) - \frac{\lambda_z(x)}{s_z}\right)^2. \tag{5}$$

*Proof.* The parameter error is

$$\sum_x \mathbb{E}[(\hat{\theta}_z(x) - p^*(x \,|\, z))^2] = \sum_x \mathbb{E}[\hat{\theta}_z(x)^2] - 2\mathbb{E}[\hat{\theta}_z(x)]p^*(x \,|\, z) + p^*(x \,|\, z)^2. \tag{6}$$

Evaluating the terms separately,

$$\mathbb{E}[\hat{\theta}_z(x)] = \frac{1}{n_z + s_z}\left[\lambda_z(x) + \sum_{i=1}^{n_z}\mathbf{1}[X_z[i] = x]\right] \tag{7}$$

$$= \frac{1}{n_z + s_z}(\lambda_z(x) + n_z p^*(x\,|\,z)) \tag{8}$$

$$\mathbb{E}[\hat{\theta}_z(x)^2] = \frac{1}{(n_z + s_z)^2}\mathbb{E}[(\lambda_z(x) + \sum_{i=1}^{n_z}\mathbf{1}[X_z[i] = x])^2] \tag{9}$$

$$= \frac{1}{(n_z + s_z)^2}\left[\lambda_z(x)^2 + 2\lambda_z(x)n_z p^*(x\,|\,z) + n_z p^*(x\,|\,z) + (n_z^2 - n_z)p^*(x\,|\,z)^2\right] \tag{10}$$

Putting it all together, the parameter error can be written as

$$\sum_x \mathbb{E}[(\hat{\theta}_z(x) - p^*(x\,|\,z))^2] = \sum_x \frac{(s_z^2 - n_z)p^*(x\,|\,z)^2 + \lambda_z(x)^2 + (n_z - 2s_z\lambda_z(x))p^*(x\,|\,z)}{(n_z + s_z)^2} \tag{11}$$

$$= \sum_x \frac{n_z p^*(x\,|\,z)(1 - p^*(x\,|\,z)) + s_z^2\left(p^*(x\,|\,z) - \frac{\lambda_z(x)}{s_z}\right)^2}{(n_z + s_z)^2} \tag{12}$$

$$= \frac{n_z H_z + s_z^2 \Delta_z}{(n_z + s_z)^2}. \tag{13}$$

$\square$

**No-tradeoff example.** Suppose there are 3 domains $z \in \{1, 2, 3\}$ and $m = 3$ vocabulary tokens $x \in \{1, 2, 3\}$. We use a symmetric Dirichlet prior (preferring a uniform token distribution) where $\lambda_z(x) = 1/3$ for all tokens $x$ and domains $z$. Here, $s_z = \sum_x \lambda_z(x) = 1$. In this setting, we show that there is a set of domain weights that has strictly lower parameter error than the baseline where we sample the same number of tokens from each domain: $n_z$ are equal for all domains $z$.

Suppose the ground truth paramaters for the unigram distributions are

$$\begin{bmatrix} 1 & 0 & 0 \\ 0.7 & 0.2 & 0.1 \\ 1/3 & 1/3 & 1/3 \end{bmatrix}, \tag{14}$$

where row $z$ contains the parameters for domain $z$. For example, token 1 has probability 1 under domain 1's unigram distribution.

For domain $z = 1$ (non-noisy domain), we have $H_1 = 0$ so the parameter error (according to Lemma D.1) is

$$\frac{s_1^2 \Delta_1}{(n_1 + s_1)^2} \tag{15}$$

which is strictly decreasing in the number of samples $n_1$.

For domain $z = 3$ (noisy domain), we have $\Delta_3 = 0$ so the parameter error is

$$\frac{n_3 H_3}{(n_3 + s_3)^2}, \tag{16}$$

by Lemma D.1. This error is minimized to zero at $n_3 = 0$ (no samples). This means that we can allocate samples elsewhere while still reducing error.

For $z = 2$ (intermediate entropy domain), we have $\Delta_2 = 0.207$ and $H_2 = 0.46$. The derivative of the parameter error with respect to the number of samples $n_2$ is

$$\frac{\partial}{\partial n_2}\frac{n_2 H_2 + s_2^2 \Delta_2}{(n_2 + s_2)^2} = \frac{H_2(s_2 - n_2) - 2s_2^2 \Delta_2}{(n_2 + s_2)^3} \tag{17}$$

which is negative when

$$n_2 > s_2 - \frac{2s_2^2 \Delta_2}{H_2}.$$
(18)

This inequality holds in this case since $\frac{2\Delta_2}{H_2} < 1$ and $s_2 = 1$. Therefore the parameter error is decreasing in the number of samples $n_2$.

Thus, any domain weights that reallocate the examples from domain 3 to domains 1 and 2 reduces the parameter error for all domains.

**What kind of domains are downweighted?**   Intuitively, we can downweight the very noisy (high entropy/difficulty) domain 3 because the initialization perfectly matches the ground truth. This allows us to reallocate samples to the other domains 1 and 2. Between these, domain 1 requires less additional samples since the parameter error decreases very quickly with the number of samples $n_1$ (the difficulty $H_1$ is zero). Thus, the easiest domains should also receive relatively less weight. In practice, positive transfer between domains (which is not captured here) can also contribute to scenarios where reweighting results in no tradeoff across domains.

**Simulation with DoReMi.**   We consider running DoReMi on the above no-tradeoff instance of the simple example with the ground truth unigram distributions in Equation 14. Note that DoReMi's domain reweighting step (Step 2, Algorithm 1) involves a loop over $T$ iterative model updates, while the estimator from Equation 2 is computed in closed form. To adapt the estimator for DoReMi, we consider an iterative version where the average is computed in an online fashion. We run DoReMi for $T = 500$ steps using minibatch size 1 over the $n = 500$ training examples with domain weight update rate $\eta = 0.5$. For the model update at step $t$ on an example $x$ from domain $z$, we increase the pseudo-count $\hat{\theta}_z(x)$ by the current domain weight $\alpha_t$ corresponding to domain $z$. Instead of using the examples in the minibatch (which is only size 1 and doesn't represent all domains), we compute the per-domain excess log-perplexities in Algorithm 1 using a fixed, independent evaluation set of 30 examples.

We compare DoReMi against a model trained with baseline domain weights, which are uniform over the 3 domains. All models are trained on $n = 500$ training examples. We evaluate the log-perplexity of a model on each domain in closed form using the ground truth unigram distribution parameters.

On this simple example, DoReMi returns domain weights $[0.39, 0.61, 0.0]$ after rounding to 2 decimal places. These weights correspond to our intuitions — the first domain (non-noisy) is increased by a small amount, the third domain (noisy) is decreased to 0 weight, and most of the weight is allocated to the second domain. We use these domain weights to generate a new dataset of 500 examples. The model trained with this new dataset improves over the baseline model in perplexity on all domains.

