# OpenReview forum: "DoReMi: Optimizing Data Mixtures Speeds Up Language Model Pretraining"
_NeurIPS.cc/2023/Conference — NeurIPS 2023 spotlight_

### Official Review · Reviewer_ajJF · 2023-07-01

**Soundness:** 3 good
**Presentation:** 3 good
**Contribution:** 3 good
**Rating:** 6
**Confidence:** 3

**Summary:**

This paper proposes to pretrain language models (LMs) by first automatically learning domain weights using a small proxy model and then pretraining a large model under the learned weights. The proposed method, Domain Reweighting with Minimax Optimization (DoReMi), improves pretraining perplexity across all domains and results in better downstream task accuracy with better efficiency.

**Strengths:**

* Originality: The paper studies an interesting direction in seeking the optimal pretraining data mixture/weighting. Both the angle and the proposed method are novel.
* Quality: The method is generally well-designed to accomplish the goal discussed and there have been plenty of experiment results that analyze the effects of DoReMi, but I feel that the paper has not sufficiently motivated the necessity and benefits of learning domain weights (see weaknesses below).
* Clarity: The paper is overall clear.
* Significance: The problem tackled in this paper (i.e., automatic data selection/weighting in pretraining) is important, and the paper shows that the method can improve pretraining perplexity and certain downstream task performance, which can be considered moderately significant. However, there are concerns regarding whether the evaluations are comprehensive w.r.t. tasks and model scales (see weaknesses below).

**Weaknesses:**

* Insufficient motivation: While it's well acknowledged that the data quality is variable across different domains, it's unclear to me whether learning domain weights and using them to construct a "better" corpus is the appropriate way. First, domains are quite coarse partitions of the data, and I'm not very convinced that assigning a single scalar weight to each domain as a whole is an ideal setup. As the authors mentioned, some data might be noisy that should be down-weighted, but shouldn't this be done at an instance level? For example, some code snippets from Github might be erroneous but others are correct. If the entire Github domain is down-weighted, it doesn't really tell apart the clean vs. noisy data, but instead puts a lower priority on learning all code-related data. Second, the authors do not seem to mention or compare with a naive baseline that directly removes noisy domains from the pretraining corpus. For example, what if only high-quality data are used? Gunasekar et al. (I'm aware that this paper came out after the paper submission deadline, but it seems relevant) showed that textbook-quality data with only 6B tokens are sufficient to train good models.
* Unclear generalization ability to large model scales: Although the authors conduct experiments across different model scales, the largest model size tested is 8B. I understand that it's very expensive to train even larger models, but for a paper studying pretraining, I'd expect to see results on larger scales (e.g., 65B), considering that the model performance has a strong correlation with model sizes. This concern appears imminent given that the proxy model does not seem to scale to larger models.
* Choice of evaluation tasks: The paper mainly evaluates on pretraining perplexity and QA-related tasks. For a paper that aims at "models that perform well on all domains", I believe the evaluation should have a more comprehensive coverage of tasks, such as coding and reasoning, especially considering that these tasks are relevant to certain domains in the pretraining corpora (e.g., Github). I'd be curious to know if the performance on code completion will be still higher than the baseline if the Github domain is down-weighted.
* (Minor) As I understand, the appendix should be submitted separately from the main paper.

Reference:
Gunasekar et al. “Textbooks Are All You Need.” 2023.

**Questions:**

* How to set the size of the proxy model given the main model size?
* How do the learned domain weights by different sizes of proxy models look like?

Please also clarify any misunderstandings in my review.

**Limitations:**

Please refer to the Weaknesses section.

---

> ### Author Rebuttal · Authors · 2023-08-10
>
> We thank the reviewer for the feedback. **ajJF notes that “the angle and the proposed method are novel” and “generally well-designed”, with “plenty of experiment results”.** We address specific questions below:
>
> > “domains are quite coarse partitions of the data, and I'm not very convinced that assigning a single scalar weight to each domain as a whole is an ideal setup. As the authors mentioned, some data might be noisy that should be down-weighted, but shouldn't this be done at an instance level?”
>
> Despite their coaseness, **we already find a significant improvement from reweighting these coarse domains, and addresses an immediate practical need for determining these domain weights**. This suggests that **a future promising direction is to define more fine-grained domains** to further increase the power of reweighting. We note this in the discussion section and will clarify in the revision.
>
> > “the authors do not seem to mention or compare with a naive baseline that directly removes noisy domains from the pretraining corpus. For example, **what if only high-quality data are used? Gunasekar et al. (I'm aware that this paper came out after the paper submission deadline, but it seems relevant) showed that textbook-quality data with only 6B tokens are sufficient** to train good models.”
>
> - **Our paper’s comparisons reflect common ways to assess the quality of domains (which is a difficult problem): 1) by intuition (The Pile) or 2) by tuning on downstream tasks (GLaM dataset).**
> - In particular, on the GLaM dataset we compare against domain weights that were tuned using the downstream tasks that we evaluate on as an oracle. Without any access to the downstream tasks, we are able to get a similar performance. **In particular, part of GLaM’s tuning process is to evaluate the performance of models trained on each single domain (including books only or web only).  Thus we believe our comparison to downstream-tuned weights is optimal, and in particular a stronger comparison than the reviewer's suggestion.**
> - Furthermore, we believe that Gunasekar et al., which came out after the paper submission deadline, can focus the pretraining dataset on (coding) textbooks since they are training more specialized models for code.
>
> > “Although the authors conduct experiments across different model scales, the largest model size tested is 8B. I understand that it's very expensive to train even larger models, but for a paper studying pretraining, I'd expect to see results on larger scales (e.g., 65B)”
>
> Although we would like to train 65B models, **due to compute limitations we instead show that DoReMi brings benefits at a variety of scales from 280M to 8B, without diminishing returns**. This suggests that DoReMi will scale well to even larger models (analogous to scaling laws).
>
> > “For a paper that aims at "models that perform well on all domains", I believe the evaluation should have a more comprehensive coverage of tasks, such as coding and reasoning … I'd be curious to know if the performance on code completion will be still higher than the baseline if the Github domain is down-weighted.”
>
> For broad evaluation, we find that **DoReMi improves perplexity on all pretraining domains**. Even though the Github domain is downweighted, we find an improvement in perplexity / next-token prediction, which is closely related to code completion and typically tracks downstream accuracy.
>
> > “How to set the size of the proxy model given the main model size?”
>
> Given that the weights are able to transfer across 30x larger model scales, we suggest **using a fixed proxy model size (e.g. 280M)** for any main model size. We thank the reviewer for the important practical question and will add a discussion in the final revision.
>
> > “How do the learned domain weights by different sizes of proxy models look like?”
>
> We compare the domain weights from 280M and 1B proxy models below. With a 280M proxy model, most of the weight is put on the Pile-CC web text domain, while DoReMi with a 1B proxy model puts most of the weight on OpenWebText2. The overall pattern of the domain weights for the rest of the domains are similar. This suggests there may be multiple local minima in domain weight space, especially when there are some similar domains (such as Pile-CC vs OpenWebText2).
>
> |                   | Baseline | DoReMi (280M) | DoReMi (1B) |
> |-------------------|---------:|--------------:|------------:|
> | Pile-CC           |   0.1121 |        0.6057 |      0.1199 |
> | PubMed Central    |   0.1071 |        0.0046 |      0.0149 |
> | Books3            |   0.0676 |        0.0224 |      0.0739 |
> | OpenWebText2      |   0.1247 |        0.1019 |      0.3289 |
> | ArXiv             |   0.1052 |        0.0036 |      0.0384 |
> | Github            |   0.0427 |        0.0179 |      0.0129 |
> | FreeLaw           |   0.0386 |        0.0043 |      0.0148 |
> | StackExchange     |   0.0929 |        0.0153 |      0.0452 |
> | USPTO Backgrounds |   0.0420 |        0.0036 |      0.0260 |
> | PubMed Abstracts  |   0.0845 |        0.0113 |      0.1461 |
> | Gutenberg (PG-19) |   0.0199 |        0.0072 |      0.0250 |
> | OpenSubtitles     |   0.0124 |        0.0047 |      0.0017 |
> | Wikipedia (en)    |   0.0919 |        0.0699 |      0.0962 |
> | DM Mathematics    |   0.0198 |        0.0018 |      0.0004 |
> | Ubuntu IRC        |   0.0074 |        0.0093 |      0.0044 |
> | BookCorpus2       |   0.0044 |        0.0061 |      0.0029 |
> | EuroParl          |   0.0043 |        0.0062 |      0.0078 |
> | HackerNews        |   0.0075 |        0.0134 |      0.0058 |
> | YoutubeSubtitles  |   0.0042 |        0.0502 |      0.0159 |
> | PhilPapers        |   0.0027 |        0.0274 |      0.0063 |
> | NIH ExPorter      |   0.0052 |        0.0063 |      0.0094 |
> | Enron Emails      |   0.0030 |        0.0070 |      0.0033 |
>
>
> > (Minor) As I understand, the appendix should be submitted separately from the main paper.
>
> We thank the reviewer for pointing this out, and will correct it in the final revision.

---

> > ### Comment · Reviewer_ajJF · 2023-08-19
> >
> > I thank the authors for their response. Some of my concerns (e.g., the omission of the naive baseline) are addressed. Although I do hope to see a finer-grained partition of the domains as well as some concrete downstream task results (instead of merely perplexity-based metrics), especially on the downweighted domains (e.g., Github), these concerns are relatively minor considering the paper's high novelty. Hence, I have updated my overall rating.

---

### Official Review · Reviewer_uiyj · 2023-07-06

**Soundness:** 4 excellent
**Presentation:** 4 excellent
**Contribution:** 4 excellent
**Rating:** 8
**Confidence:** 4

**Summary:**

This paper introduces a significant advancement by exploring the topic of data mixture proportions during pre-training, which holds great importance. Determining how to sample pre-trained data from diverse sources to achieve balanced results is a fundamental question. Previous approaches have often relied on intuitive-based weights or extensive experimentation to select a setting. However, these approaches either require extensive computational resources or lack generalization across different settings. Therefore, finding the optimal mixture settings using a smaller model is an intriguing and crucial question.

To address this, the authors propose a group-DRO-based method for determining the weights. The weight is dynamically adjusted during the learning process, and the final weight is selected as the sampling weight. Additionally, the authors conducted extensive experiments, providing substantial evidence to support the effectiveness of the proposed approach.

**Strengths:**

The direction is really important and the general framework is valuable that uses a small network to get the best practice and apply it on larger models.


The proposed method is clearly-written, and the improvements it offers have been convincingly substantiated through extensive experiments.

The general idea is also very elegant. If my understanding is correct, the method generally assigns larger weights to the data that can be learned in the future, and assigns smaller weights to those too easy or difficult data. The general idea is cool.

**Weaknesses:**

The selected weights vary significantly across different models with varying scales.

I still have some concerns about the final implementation. Following the optimization objectives, the authors use the learned weight as the re-sampling weight. It is a little bit strange.

**Questions:**

For instance, the selection of the best weight differs among models with different scales. Does this imply the existence of multiple sub-optimal weight candidates? Moreover, I still do not comprehend why the authors do not apply the final model during the learning process but instead utilize the final weights for re-weighting purposes.

---

> ### Author Rebuttal · Authors · 2023-08-10
>
> We thank the reviewer for the feedback. Overall, uiyj feels that the “the direction is really important”, “the general framework is valuable”, and “the improvements it offers have been convincingly substantiated through extensive experiments”. We address specific questions below:
>
> > “the selection of the best weight differs among models with different scales. Does this imply the existence of multiple sub-optimal weight candidates?”
>
> **We believe that there is a frontier of solutions, especially when there are domains that are similar (e.g., OpenWebText and Pile-CC).** For example, we compare the domain weights from 280M and 1B proxy models below. With a 280M proxy model, most of the weight is put on the Pile-CC web text domain, while DoReMi with a 1B proxy model puts most of the weight on OpenWebText2. The overall pattern of the domain weights for the rest of the domains are similar. We thank the reviewer for the question and will include this discussion in the final revision.
>
> |                   | Baseline | DoReMi (280M) | DoReMi (1B) |
> |-------------------|---------:|--------------:|------------:|
> | Pile-CC           |   0.1121 |        0.6057 |      0.1199 |
> | PubMed Central    |   0.1071 |        0.0046 |      0.0149 |
> | Books3            |   0.0676 |        0.0224 |      0.0739 |
> | OpenWebText2      |   0.1247 |        0.1019 |      0.3289 |
> | ArXiv             |   0.1052 |        0.0036 |      0.0384 |
> | Github            |   0.0427 |        0.0179 |      0.0129 |
> | FreeLaw           |   0.0386 |        0.0043 |      0.0148 |
> | StackExchange     |   0.0929 |        0.0153 |      0.0452 |
> | USPTO Backgrounds |   0.0420 |        0.0036 |      0.0260 |
> | PubMed Abstracts  |   0.0845 |        0.0113 |      0.1461 |
> | Gutenberg (PG-19) |   0.0199 |        0.0072 |      0.0250 |
> | OpenSubtitles     |   0.0124 |        0.0047 |      0.0017 |
> | Wikipedia (en)    |   0.0919 |        0.0699 |      0.0962 |
> | DM Mathematics    |   0.0198 |        0.0018 |      0.0004 |
> | Ubuntu IRC        |   0.0074 |        0.0093 |      0.0044 |
> | BookCorpus2       |   0.0044 |        0.0061 |      0.0029 |
> | EuroParl          |   0.0043 |        0.0062 |      0.0078 |
> | HackerNews        |   0.0075 |        0.0134 |      0.0058 |
> | YoutubeSubtitles  |   0.0042 |        0.0502 |      0.0159 |
> | PhilPapers        |   0.0027 |        0.0274 |      0.0063 |
> | NIH ExPorter      |   0.0052 |        0.0063 |      0.0094 |
> | Enron Emails      |   0.0030 |        0.0070 |      0.0033 |
>
>
> >  “I still do not comprehend why the authors do not apply the final model during the learning process but instead utilize the final weights for re-weighting purposes.”
>
> **We do not use DRO to directly train the large model because it is a more expensive training procedure** that evaluates the losses of two equally sized models (proxy and reference) at every training step. **Instead, we do the expensive DRO training at a small scale, transferring the benefits to the large model through the optimized domain weights.** This also preserves the standard training procedure for the large model. We apologize for the confusion and will clarify in the final revision.

---

> > ### Comment · Reviewer_uiyj · 2023-08-20
> >
> > Thanks for the response. I will keep my score.

---

### Official Review · Reviewer_5e3R · 2023-07-06

**Soundness:** 4 excellent
**Presentation:** 4 excellent
**Contribution:** 4 excellent
**Rating:** 7
**Confidence:** 5

**Summary:**

The authors proposed DoReMi for optimizing the mixture proportions of pretraining data domains when training language models (LMs). The authors demonstrate that DoReMi, which utilizes a small proxy model trained via group distributionally robust optimization (Group DRO), can be used to determine optimal domain weights without knowledge of downstream tasks. Subsequently, these weights are used to resample a dataset for training a larger model. Experimental results indicate significant improvements in LM performance using DoReMi, including a 6.5% increase in average few-shot downstream accuracy and achieving baseline accuracy with 2.6x fewer training steps.

**Strengths:**

- DoReMi offers a unique and efficient way to determine domain weights, speeding up LM training and improving accuracy. Moreover, the down-weighted domains are not getting a worse performance, which is surprising.
- It also shows that the domain weights determined by DoReMi are transferable across a broad range of model scales, compute budgets, and other training hyperparameters, making it wide applicability.
- Overall, the idea of DoReMi is simple, and the results can show its effectiveness.

**Weaknesses:**

- Lack of the baselines: In this paper, the baselines are the LMs trained on the original data distribution of Pile, but there should be some simple but stronger baselines as well, like calculating the lexical overlap within each domain and assigning weights to maximize the lexical diversity. Although this simple baseline sounds naive, we still need to justify that their improvement would not be as much as DoReMi.
- DoReMi needs two or more proxy LM training processes in order to obtain the domain weights. However, the authors need to justify that there doesn't exist simpler but equally effective heuristics that only need one proxy LM training, such as the number of example forgetting times [1].
- Based on Figure 6, it seems that the choice of proxy model size is very crucial. For 8B model, the 280M proxy is significantly better than other larger or smaller proxy models. It would be better if the author can provide explanation or principles to select the size of the proxy model, otherwise, people who want to use DoReMi actually need to empirically run different proxy models in order to get better performance.

[1] An Empirical Study of Example Forgetting during Deep Neural Network Learning. Mariya Toneva, Alessandro Sordoni, Remi Tachet des Combes, Adam Trischler, Yoshua Bengio, Geoffrey J. Gordon. ICLR 2019
https://arxiv.org/abs/1812.05159


**Questions:**

- For the domain weight $\alpha$, I am wondering why we need gradient ascent to update it? Isn't it already optimal if we simply assign 1.0 for the domain with the highest excess loss and 0.0 for all the other domains? If the reason is that we need a single set of consistent domain weights till the end of the training, we could also collect the weights along the training process and average them at the end. Did you try this before?
- I am curious whether there are simple heuristics that can achieve the same level of effects. For example, in [1], people found that the number of forgetting times of a training example is a good indicator of how important each training example is. We need this kind of baseline to prove that the 2 runs in DoReMi are necessary and its result is more effective than others.

[1] An Empirical Study of Example Forgetting during Deep Neural Network Learning. Mariya Toneva, Alessandro Sordoni, Remi Tachet des Combes, Adam Trischler, Yoshua Bengio, Geoffrey J. Gordon. ICLR 2019
https://arxiv.org/abs/1812.05159


**Limitations:**

Yes.

---

> ### Author Rebuttal · Authors · 2023-08-10
>
> We thank the reviewer for the feedback. 5e3R notes that “DoReMi offers a unique and efficient way to determine domain weights” with “wide applicability”. We address specific questions below:
>
>
> > “the baselines are the LMs trained on the original data distribution of Pile, but there should be some simple but stronger baselines as well, like calculating the lexical overlap within each domain and assigning weights to maximize the lexical diversity.”
>
> - Our paper’s comparisons reflect common ways to assess the quality/importance of domains: 1) by intuition (The Pile) or 2) by tuning on downstream tasks (GLaM dataset).
> - In particular, on the GLaM dataset, we compare against domain weights that were tuned using the downstream tasks that we evaluate on as an oracle. Without any access to the downstream tasks, we are able to get a similar performance. **We believe the comparison to downstream-tuned weights is optimal and in particular, stronger than the simple baseline proposed by the reviewer.**
>
> > “DoReMi needs two or more proxy LM training processes in order to obtain the domain weights. However, the authors need to justify that there doesn't exist simpler but equally effective heuristics that only need one proxy LM training, such as the number of example forgetting times [1].”
>
> - Overall, the amount of compute used to train a small proxy model is much smaller than the compute needed to train a large model (2.5% in our 280M to 8B experiment), so that **saving compute in the small proxy model training step only results in a small compute benefit.**
> - **In many practical settings, there may already exist a pretrained reference model** that can be used, reducing the number of proxy LM training processes to 1.
>
> > “It would be better if the author can provide explanation or principles to select the size of the proxy model, otherwise, people who want to use DoReMi actually need to empirically run different proxy models in order to get better performance.”
>
> - We believe that we can use a fixed proxy model size (e.g. 280M) to find the domain weights for any main model size, given that the weights are able to transfer across 30x larger model scales. We thank the reviewer for the important practical question and we will add more discussion on this in the final revision.
>
> > “For the domain weight α, I am wondering why we need gradient ascent to update it? Isn't it already optimal if we simply assign 1.0 for the domain with the highest excess loss and 0.0 for all the other domains?”
>
> Although assigning all the weight to the domain with the highest excess loss is optimal for the inner maximization, **it can result in unstable training for the outer minimization** since it reduces the number and diversity of examples in the minibatch. Instead, we follow Sagawa et al. (https://arxiv.org/abs/1911.08731) and employ a mirror descent-based DRO optimizer that updates the domain weight α smoothly while maintaining optimality guarantees.

---

> > ### Comment · Reviewer_5e3R · 2023-08-16
> > **Thanks for the responses**
> >
> > The responses well addressed my questions. I would keep my score of 7 Accept.

---

### Official Review · Reviewer_VPSZ · 2023-07-08

**Soundness:** 3 good
**Presentation:** 4 excellent
**Contribution:** 4 excellent
**Rating:** 8
**Confidence:** 5

**Summary:**

This paper introduces DoReMi, a method for automatically deriving optimal/improved domain weights for aggregated pretraining datasets for LLMs. DoReMi works in three 3: (1) train a small reference model to use for the excess loss in DRO; (2) train a small proxy model with DRO to obtain optimised domain weights; (3) use the optimised weights with a larger model. The authors demonstrate their method leads to significant improvements on The Pile, and also matches manual tuning on GLaM. In ablations, the scaling behaviour is studied, showcasing that proxy model size improves downstream performance up to ~280M parameters, after which improvements degrade as the larger models are inadequately trained by DRO.

**Strengths:**

* **S1.** Automatic optimisation of domain weights for pretraining datasets is an incredibly valuable contribution for the LLM community. Standard procedures have been mostly based on expensive manual tuning, and have not necessarily been principled in their finding.

* **S2.** The paper is well grounded in works around DRO, and adequately position itself -- diverging from existing methods when necessary. The authors also take care in pointing out current limitations of the DRO approach, and potential for future improvements.

* **S3.** The results obtained on The Pile reproduce the observations recently made by the RedPjamas & RefinedWeb datasets: some components of The Pile should ideally be downsampled, and increased web data may be beneficial. The fact that DoReMi repeatedly reproduces results obtained from manual tuning is a good validation of the method.

* **S4.** The paper is well-written and presented, and easy to follow.

**Weaknesses:**

* **W1. The evaluation setup could be broader.** The authors evaluate downstream performance: (1) in 1-shot; (2) in a generative/exact-match setting; (3) on 5 tasks.
   * **W1.1.** The choice of a generative exact-match setting is strange for models in the ~100M-1B range, as they consistently struggle with exact match. Instead, for small models, leveraging logprob-based evaluation of multiple choices is more common, and can deliver stronger signal.
   * **W1.2.** The choice of task is arbitrary. Rather than these 5 tasks, the authors could have evaluated on the full set of GPT-3 tasks, or on a popular benchmark such as HELM or BigBench (for the larger models).

* **W2. The poor scaling behaviour of DoReMi past 280M parameters for the proxy model is a concern for robustness.** Notably, this might make it difficult for practitioners to apply DoReMi as a "set it and forget it" method. Some level of manual inspection & analysis is required, which holds back the method from fully delivering on its promise of completely automated domain weight optimization. However, I appreciate that the authors discuss this limitation openly and propose some potential ideas for further improvements in this direction.

* **W3.** (smaller nits)
    * **W3.1.** The authors showcase in Table 1 the baseline & DoReMi domain weights on The Pile; the presentation of the table could be improved, to better identify which domains are upsampled and which are downsampled -- this could be as simple as sorting the domains and explicitly providing the up/downsampling value in the table.

**Questions:**

This is an excellent paper introducing a method which could see wide adoption in the community as a way to improve aggregated pretraining datasets. The paper is well-written, and opens the door to numerous exciting follow-up works. Accordingly, I would currently rate it as a **Strong Accept (8)**. Note that should my concerns about evaluation be addressed, I would be willing to further increase my score to a 9/10 -- this paper has significant potential for the community, and my only main concern currently is regarding the robustness of the selected evaluation setup.

* **Q1.** (W1.1.) Could the authors better justify their choice of going with generative exact match for the evaluation?

* **Q2.** (W1.2.) Could the authors provide scores (in 0-shot or 1-shot) using logprob-based multiple choices instead?

* **Q3.** (W1.2.) Could the authors provide evaluation results on additional tasks, such as the full set of GPT-3 tasks, or scores on a popular benchmark such as HELM or BigBench? (this applies mostly to the headline 8B models).

**Limitations:**

The authors extensively discuss limitations of their work in a dedicated section and provide interesting pointers for further research.

---

> ### Author Rebuttal · Authors · 2023-08-10
>
> We thank the reviewer for the feedback. VPSZ felt that the method “is an incredibly valuable contribution”, “reproduces results obtained from manual tuning” from the community, and “could see wide adoption”. We address specific questions below:
>
>
> > “The choice of a generative exact-match setting is strange for models in the ~100M-1B range, as they consistently struggle with exact match. Instead, for small models, leveraging logprob-based evaluation of multiple choices is more common, and can deliver stronger signal… Rather than these 5 tasks, the authors could have evaluated on the full set of GPT-3 tasks, or on a popular benchmark such as HELM or BigBench (for the larger models).”
>
> **We evaluated on generative exact-match tasks since that is how the models are typically used** (prompting, rather than scoring logprobs). We note that the GPT-3 paper finds that performance on TriviaQA, for example, scales smoothly with model size even at the 100M-1B range, and tracks other benchmarks well. For broad evaluation, we show that DoReMi improves the validation perplexity on all domains, which typically has a strong relationship with downstream performance.
>
>
> > “The poor scaling behaviour of DoReMi past 280M parameters for the proxy model is a concern for robustness.”
>
> VPSZ is referring to the scaling behavior of DoReMi with respect to proxy model size, where larger (1B) proxy models were found to be less well optimized with DRO. Overall, **we saw gains across all the proxy model sizes**, but we agree with VPSZ that improving the DRO optimization for larger proxy models is an important future step.
>
> > Nits: table presentation
>
> We thank the reviewer for the suggestions, and will include them in the final revision.

---

> > ### Comment · Reviewer_VPSZ · 2023-08-16
> > **Answer to rebuttal**
> >
> > First, I would like to thank the authors for taking time to write a rebuttal to each reviewer.
> >
> > Based on the rebuttal and the other reviews, I will maintain my score of a **Strong Accept (8)**.

---

### Official Review · Reviewer_MeFt · 2023-07-20

**Soundness:** 2 fair
**Presentation:** 4 excellent
**Contribution:** 3 good
**Rating:** 7
**Confidence:** 4

**Summary:**

This work buils upon prior studies’ empirical findings, emphasizing how the composition of pretraining data affects the performance of Language Models (LMs). To avoid reliance on heuristic or iterative performance measurements on downstream tasks, this work introduces a method that employs a trainable model, capable of assigning appropriate weights to each “domain” of pretrainng data.

The proposed approach leverages the concept of Distributionally Robust Optimization (DRO) to train a small proxy model that learns to assign weights to each domain, thereby minimizing the worst-case excess losses. Subsequently, a larger model is trained using pretraining data that has been adjusted by these optimized domain weights.

The authors demonstrate through experiments that their proposed method significantly accelerates the pretraining process by showing the few-shot accuracy on factual QA tasks. They show that the model trained with optimized pretraining data reaches the performance level of baseline models much faster.

**Strengths:**

- The idea to optimize the pre-training data sounds clever, particularly their use of a proxy model trained with DRO. This offers a novel approach to learning the optimal data distribution without the costly estimation of the large pre-trained model’s task performance under different pretraining data settings.
- This work provides solid experimental evidence, exploring a variety of settings including various proxy and main model sizes, two widely-used pretraining datasets (GLaM dataset and The Pile), and different domain compositions. These experiments adequately address questions that arise during a review of the paper to some extent.
- This research yields some interesting observations. According to Tables 1 and 2, the webpage domain carries the largest weights across both pretraining datasets. This suggests that optimizing the model with respect to the webpage domain could potentially reduce excess loss in other domains. Consequently, pretraining large models with an optimized dataset results in lower perplexities across all domains. Interestingly, despite a decrease in weight for Wikipedia, task performance on tasks derived from it (TriviaQa, NaturalQuestions) improves (Table 5 in Appendix).
These insights could prove beneficial for other researchers in the field.

**Weaknesses:**

- The diversity of downstream tasks examined is relatively narrow, given the paper’s claim.
While the experiments in this paper present solid empirical evidence that the proposed method improves language model performance, the evidence primarily focuses on language modeling and factual question answering tasks in a general domain. As such, the claim that the proposed method “speeds up language model pretraining” might mislead readers, since the evaluation of pretraining should be more rigorous. For example, incorporating experimental results on MMLU [1] or commonsense reasoning (ARC[2], CSQA[3], etc) could increase the credibility of the claim that the pre-trained model with the proposed method generally outperforms previous approaches.

[1] Measuring Massive Multitask Language Understanding

[2] Think you have Solved Question Answering? Try ARC, the AI2 Reasoning Challenge

[3] Complex Sequential Question Answering: Towards Learning to Converse Over Linked Question Answer Pairs with a Knowledge Graph

**Questions:**

1) Why did the authors only experiment with a 280M main model in Figure 6 (right)? I understand the potential cost issues, but I believe it’s essential to show that DoReMi outperforms the webpage-only setting, even with a larger model.

2) Could you clarify how the main model employs domain weight in detail (Step 3 in Section 2)? I was unable to find detailed information on this in the paper. Does the model sample instances in the mini-batch according to the domain weight during pre-training?

Note: The appendix should be submitted as the separated supplementary file.

**Limitations:**

The authors discuss limitations in Section 6. However, the limitations regarding the evaluation methods for pretraining are not thoroughly addressed in the paper. Please refer to the “Weaknesses” section for more details on my perspective.

---

> ### Author Rebuttal · Authors · 2023-08-10
>
> We thank the reviewer for the feedback. MeFt felt that the paper presented a “novel approach to learning the optimal data distribution”, provides “solid experimental evidence, exploring a variety of settings”. We address specific questions below:
>
> > “While the experiments in this paper present solid empirical evidence that the proposed method improves language model performance, the evidence primarily focuses on language modeling and factual question answering tasks in a general domain…For example, incorporating experimental results on MMLU [1] or commonsense reasoning (ARC[2], CSQA[3], etc) could increase the credibility”
>
> We evaluated on generative exact-match tasks since that is how the models are typically used (prompting, rather than scoring logprobs). We note that the GPT-3 paper finds that performance on TriviaQA, for example, scales smoothly with model size even at the 100M-1B range, and tracks other benchmarks well. For broad evaluation, we show that DoReMi improves the validation perplexity on all domains, which typically has a strong relationship with downstream performance.
>
> > “ Why did the authors only experiment with a 280M main model in Figure 6 (right)? I understand the potential cost issues, but I believe it’s essential to show that DoReMi outperforms the webpage-only setting, even with a larger model.”
>
> - We conducted the ablations of the excess loss objective in Fig 6 using 280M models due to the high cost of running large-scale experiments.
> - With regards to the webpage-only comparison: On the GLaM dataset, we compare against GLaM domain weights that were tuned using the downstream tasks that we evaluate on as an oracle. Without any access to the downstream tasks, we are able to get a similar performance. **Part of GLaM’s process of tuning against downstream tasks is to evaluate the performance of models trained on each single domain. Thus we believe the comparison to downstream-tuned weights is a stronger comparison than the webpage-only baseline proposed by the reviewer.** We thank the reviewer for the question and will clarify this in the final revision.
>
> > “Could you clarify how the main model employs domain weight in detail (Step 3 in Section 2)? … Does the model sample instances in the mini-batch according to the domain weight during pre-training?”
>
> The main model trains in a standard way with data sampled from the optimized domain weights. As the reviewer mentions, the data loader samples instances from each domain according to the domain weights, and then combines these instances into a minibatch on-the-fly during pre-training. This can be implemented efficiently with separate data queues for each domain. By sampling on the fly, the reweighted dataset never has to be materialized. We apologize for any confusion and will clarify in the final revision.
>
> > “Note: The appendix should be submitted as the separated supplementary file.”
>
> We thank the reviewer for the note and will update this in the final revision.

---

> > ### Comment · Reviewer_MeFt · 2023-08-17
> >
> > Thank you for the rebuttal. I will maintain my score.

---

### Decision · Program_Chairs · 2023-09-21

**Decision:**

Accept (spotlight)

**Comment:**

This paper proposes a novel effective domain-reweighting method for LLM pretraining. The topic is very interesting and significant.

All reviewers appreciated for the contributions and gave high acceptance scores.

AC also agrees the reviewers' thoughts, so recommends accepting this paper as spotlight or oral